# Salivary Interleukin Levels in Oral Squamous Cell Carcinoma and Oral Epithelial Dysplasia: Findings from a Sri Lankan Study

**DOI:** 10.3390/cancers15051510

**Published:** 2023-02-28

**Authors:** Nadisha S. Piyarathne, Manjula M. Weerasekera, Pasquel Fonsekalage Damith Fonseka, Appu Hennedi Thotahewage Sunil Karunatilleke, Rubasinha Liyanage Pemith Ranura Liyanage, Ruwan Duminda Jayasinghe, Kanishka De Silva, Surangi Yasawardene, Ekta Gupta, Jayasinghe Arachchilage Premasiri Jayasinghe, Rasha Abu-Eid

**Affiliations:** 1Institute of Dentistry, School of Medicine, Medical Sciences and Nutrition, University of Aberdeen, Aberdeen AB25 2ZR, UK; 2Center for Research in Oral Cancer, Faculty of Dental Sciences, University of Peradeniya, Peradeniya 20400, Sri Lanka; 3Faculty of Medical Sciences, University of Sri Jayewardenepura, Nugegoda 10250, Sri Lanka; 4Sri Lanka Institute of Biotechnology, Pitipana, Homagama 10206, Sri Lanka; 5Colombo South Teaching Hospital, Kalubowila 10350, Sri Lanka; 6National Dental Hospital (Teaching), Colombo 00700, Sri Lanka; 7National Cancer Institute, Maharagama 10280, Sri Lanka; 8Aberdeen Cancer Centre, University of Aberdeen, Aberdeen AB25 2ZR, UK

**Keywords:** saliva, biomarkers, oral squamous cell carcinoma, oral epithelial dysplasia, interleukins

## Abstract

**Simple Summary:**

The incidence of oral cancer is increasing with little improvement in survival. This is due to late diagnosis with most cases diagnosed at a stage beyond a cure. In Sri Lanka, oral cancer is the most common cancer in males, carrying a high mortality rate. Identifying markers that can help in early detection is important for improving patient outcome. Here, we present findings from a study in Sri Lanka that assessed salivary interleukins in oral cancer and precancer patients compared to disease-free controls. Our results clearly show that the salivary protein levels of these interleukins progressively increase from disease-free participants through different grades of dysplasia with the highest levels reported in cancer patients. This has significant clinical potential, as assessment of salivary levels of these interleukins can be developed into non-invasive risk assessment tools for detecting disease progression in dysplasia, and as screening tests for early detection of oral cancer.

**Abstract:**

The incidence of oral squamous cell carcinoma (OSCC), and its precursor, oral epithelial dysplasia (OED), is on the rise, especially in South Asia. OSCC is the leading cancer in males in Sri Lanka, with >80% diagnosed at advanced clinical stages. Early detection is paramount to improve patient outcome, and saliva testing is a promising non-invasive tool. The aim of this study was to assess salivary interleukins (lL1β, IL6, and IL8) in OSCC, OED and disease-free controls in a Sri Lankan study cohort. A case-control study with OSCC (n = 37), OED (n = 30) patients and disease-free controls (n = 30) was conducted. Salivary lL1β, IL6, and IL8 were quantified using enzyme-linked immuno-sorbent assay. Comparisons between different diagnostic groups and potential correlations to risk factors were assessed. Salivary levels for the three tested interleukins increased from disease-free controls through OED, and were highest in OSCC samples. Furthermore, the levels of IL1β, IL6, and IL8 increased progressively with OED grade. The discrimination between patients (OSCC and OED) and controls, as assessed by AUC of receiver operating characteristic curves, was 0.9 for IL8 (*p* = 0.0001) and 0.8 for IL6 (*p* = 0.0001), while IL1β differentiated OSCC from controls (AUC 0.7, *p* = 0.006). No significant associations were found between salivary interleukin levels and smoking, alcohol, and betel quid risk factors. Our findings suggest that salivary IL1β, IL6, and IL8 are associated with disease severity of OED, and are potential biomarkers for predicting disease progression in OED, and the screening of OSCC.

## 1. Introduction

Oral cancer is among the top twenty cancers in the world [1] with more than 90% of malignancies being oral squamous cell carcinomas (OSCCs) [2]. In the year 2020, more than 377,000 new cases of lip and oral cavity cancers were reported globally [1], with more than two-thirds of the cases reported in low- and middle-income countries in Asia, where there is an increasing trend in incidence rates [3]. In Asia, the main risk factors of OSCC are betel quid chewing, smoking and alcohol, in the backdrop of poor oral hygiene, malnutrition, and limited healthcare [4].

Sri Lanka is an island nation, where lip and oral cavity cancer is the number one cancer in the male population, with 20.4 crude incidence rate, accounting for more than 2000 new cases a year [5]. The main risk factor is betel quid chewing, where the risk is elevated when combined with smoking and alcohol [6]. Even though tobacco cessation programs have led to declining OSCC incidence over the past decades [7], Sri Lanka still reports one of the highest age-standardized incidence rates of OSCC in Asia [8], where more than 80% of OSCC cases are diagnosed at advanced clinical stages [9].

Diagnosis of OSCC at advanced clinical stages leads to poor prognosis and significant complications, including treatment failure, toxicities, side effects due to radio- and chemotherapy, and complications due to the altered oral microbiome [10,11,12]. Despite advances in treatment modalities, OSCC has a low five-year survival rate, and there was no significant improvement in the survival rates between 1980 and 2010 [13]. The health cost involved in managing advanced-stage OSCC patients was five- to six-fold higher compared to the patients diagnosed at early stages [14,15]. Low survival is mainly due to delayed diagnosis, and a lack of screening protocols was highlighted as a major barrier for the early detection of OSCC [16].

Oral epithelial dysplasia (OED) is the histological presentation that precedes many OSCCs. OED is characterized by cellular and architectural abnormalities, where lesions are categorized as mild, moderate, and severe, or high-risk and low-risk grades [17]. Adjuvant techniques to aid risk assessment of OED, using molecular markers, are gravely needed [18]. These could help in predicting disease progression of OED, thus aiding early detection of malignant transformation.

Saliva testing is a suitable tool for the early detection of OSCC, due to proximity of the sample to the diseases tissue [19]. Saliva is rich in biomolecules and different constituents, such as cytokines, micro-RNAs and metabolites, identified as biomarkers for OSCC in saliva [20,21,22,23]. The advantages of saliva over conventional samples, such as blood and biopsy, are non-invasiveness, safe handling and transport, possibility of repeated sampling, and suitability for community screening and chairside detection tests.

Interleukins (ILs) are secreted by immune cells, tumour-associated macrophages, and cancer cells, among others [24], and are associated with the pathogenesis of OSCC [25]. Among those, IL6 has both pro- and anti-inflammatory properties, and pro-inflammatory signals are predominant in the pathogenesis of cancer [26]. IL6, at both protein and RNA levels, was higher in patients compared to controls, and was identified as a robust biomarker for OSCC [27]. Another IL present in the tumour microenvironment is IL8. It is associated with tumour genesis, angiogenesis, and metastasis [28]. IL1β is an IL predominant in chronic inflammation and it promotes tumour initiation and progression [29]. Significant differences in salivary IL1β, IL6, and IL8 were reported in patients with OSCC compared to disease-free controls [30,31,32,33]. Recent systematic reviews and meta-analyses compile evidence that IL1β, IL6, and IL8 are potential salivary biomarkers for the early detection of OSCC [20,34,35,36].

Disease-specific salivary biomarkers need to be population-tailored with the determination of threshold values, as variable expressions in different geographies were observed [37,38]. This variation may be due to genetic, epigenetic, environmental, and associated habit-related factors. The aim of this study was to investigate the levels of IL1β, IL6, and IL8 in saliva in patients with OSCC and OED compared to disease-free controls, and their associations with risk factors in a previously unreported, Sri Lankan population.

## 2. Methods

### 2.1. Ethical Approvals

This study adhered to ethical guidelines specified by the declaration of Helsinki. Ethical approvals and written permissions for the study were obtained from the College Ethics Review Board, School of Medicine, Medical Sciences and Nutrition, University of Aberdeen, United Kingdom (Ref: CERB/2019/8/1821); Ethics Review Committee, Faculty of Medical Sciences, University of Sri Jayewardenepura, Sri Lanka (Ref: 21/19); and Education, Training and Research division, Ministry of Health, Sri Lanka (Ref: ETR/AC/2019/11).

### 2.2. Recruitment of Study Participants

Study participants were recruited from three hospitals in Sri Lanka (Colombo South Teaching Hospital, National Cancer Hospital and The Family Practice Centre of the University of Sri Jayewardenepura). For the OSCC and OED groups, patients with confirmed histopathological diagnoses of OSCC or OED (with biopsy reports issued by the participating institutes) were selected. For disease-free controls, volunteers who did not have a clinical or histopathological diagnosis of OSCC, or oral potentially malignant disorders, were selected. All participants were adults (>18 years). Volunteers who could not give informed written consent, were pregnant and lactating mothers, patients with immunosuppression and subjects who suffered at the time, or previously from any malignancy, were excluded. The control group was age- and sex-matched to the patients. Informed written consent was obtained from all participants. Data on socio-demographic details (age, gender, income, occupation) and risk factors (smoking, alcohol, and betel quid) were obtained using an interviewer-administered questionnaire. Medical history was obtained from hospital records. Recruitment, data, and sample collection were conducted between 1 July 2019 and 1 March 2020.

### 2.3. Saliva Sample Collection and Processing

Unstimulated whole saliva samples were obtained from participants between 9.00 am and 12.00 pm, to avoid diurnal variation. Donors were asked to refrain from eating, tooth brushing, mouthwash use, smoking, betel quid chewing and alcohol intake for at least 2 h prior to sample collection. At the saliva collecting appointment, donors were asked to wash their mouth with drinking water and discard all the contents in the mouth, rest for 5 min with the head tilted forward and without swallowing, to allow for saliva collection. Following that, they were instructed to spit out the collected oral fluid (2–5 mL) into a sterile plastic falcon tube (15 mL) on ice. Samples with visible blood contamination were excluded. Samples were transported on ice and then centrifuged at 2600 rpm for 15 min at 4 °C. The supernatants were separated, and 50X protease inhibitor cocktail (Promega USA-G6521) was added, and the samples were stored at −70 °C until analysis.

### 2.4. Interleukin Quantification

Protein levels of ILs were quantified using a commercially available sandwich enzyme-linked immune-sorbent assay (ELISA), using pre-coated plates: E-EL-H0149, E-EL-H0102 and E-EL-H0048 kits (Elabscience; Wuhan, Hubei, China). The detection range for all three ELISA kits was 7.8–500 pg/mL. Following optimization, a 1:10 dilution was used for the OSCC group. The optical density (OD) values at 450 nm were measured using a spectrophotometer (Multiskan SkyHigh with a cuvette, Thermo Fisher Scientific), with duplicate sample testing. For samples exceeding the OD of the highest standard, the measurement was repeated with a 1:10 dilution. Calculations were done by interpolation against an eight-point standard curve for each plate, using the Graph Pad Prism (version 6) software.

### 2.5. Statistical Analysis

Data were anonymized and entered to SPSS 20 for windows (IMB SPSS statistics, IMB Cooperation, New York, NY, USA), and analysed using SPSS 20 for windows and Graph Pad Prism 6. Data were not normalised or transformed. The normality of the distribution was assessed using Shapiro–Wilk normality test, and the salivary biomarker data deviated from the normal distribution, therefore non-parametric tests were used. Groups were compared using the Chi-square test, Kruskal Wallis test, and the Dunn’s test. The receiver operating characteristic (ROC) curves were used to assess the discrimination between three distinctions (OSCC vs. controls, OED vs. controls and OSCC vs. OED), and the cut-off values for the ROC curves were determined by identifying the point with high specificity (>80%) and related maximum sensitivity [39]. *p*-values less than 0.05 were considered statistically significant.

## 3. Results

### 3.1. Study Cohort

Salivary interleukins were quantified in 97 subjects (OSCC n = 37, OED n = 30 and controls n = 30). The socio-demographic characteristics and risk factors of the study cohort are presented in Table 1. There were no statistically significant differences in the mean age, sex, and ethnicity between the study groups (*p* > 0.05). The control group had a higher income compared to the disease groups, indicating lower socioeconomic status of OSCC and OED patients. A chi-square test revealed that OSCC and OED groups had a higher percentage of betel quid chewers compared to controls (*p* = 0.001). In the OED group, there was a higher percentage of mouthwash users compared to OSCC and controls (*p* = 0.005), while the OSCC group had a significantly lower percentage of participants with co-morbidities compared to OED and controls (*p* = 0.013). There was no significant difference in other risk factors or demographics among study groups (Table 1).

The OSCC cases included well-differentiated (n = 14, 38%), moderately differentiated (n = 18, 49%), and poorly differentiated (n = 4, 10%) OSCCs, and in one patient, differentiation was not reported. The OED patients included mild dysplasia (n = 15, 50%), moderate dysplasia (n = 10, 34%) and severe dysplasia/carcinoma in situ (n = 5, 16%). Most lesions were localized in the buccal mucosa and tongue. The anatomical locations of the lesions are presented in Table 2.

### 3.2. Salivary Interleukins at the Protein Level

From the 97 study participants, 74 samples were tested for IL1β, 97 for IL6, and 93 samples for IL8. Raw data for the interleukin values (ELISA) are presented in Appendix A. Samples from OSCC and OED patients had a significantly higher level of all three tested ILs in comparison to the disease-free controls (Figure 1). IL6 and IL8 were significantly higher in OSCC in comparison to OED (all grades combined), but none of the biomarkers reported had significant differences between OED and the disease-free control samples (Figure 1). When looking at different grades of OED, a similar pattern was observed, especially for IL6 and IL8, where the biomarkers progressively increased from healthy, mild dysplasia, moderate dysplasia, and severe dysplasia, with the highest levels observed in OSCC (Figure 2). Figure 3 shows a comparison of interleukin levels in well-differentiated, moderately differentiated and poorly differentiated OSCC. Our results did not reveal a consistent pattern or statistically significant association between different OSCC grades.

The discriminating ability of the biomarkers was assessed using receiver operating characteristic (ROC) curves (Figure 4, Table 3). According to data in Table 3, IL6 demonstrated 80% (AUC = 0.8) ability and IL8 demonstrated 90% (AUC = 0.9) ability to differentiate OSCC from both OED and control groups. IL1β showed an AUC of 0.72 to differentiate OSCC from controls. Cut-off values for biomarkers were calculated to maximize specificity and sensitivity (Table 3).

Salivary IL1β, IL6, and IL8 levels were compared between participants with and without risk factors in the three study groups. In this analysis, smoking, alcohol, betel quid, and family history of cancer did not show any significant pattern of associations (data reported in Appendix A).

## 4. Discussion

Oral squamous cell carcinoma poses a significant economic burden to low- and middle-income countries in South Asia. Biomarkers related to the disease progression in OED, and screening of OSCC, are paramount to reduce the disease-associated morbidity and mortality through early detection.

Molecular changes precede histopathological and clinical presentations in cancer. The “omics” approaches in discovering biomarkers have revealed genomic, proteomic, transcriptomic, and metabolomic biomarkers in OSCC [40,41]. Many of these biomarkers were associated with the progression of precursor lesions into OSCC [18]. Recently, genes related to cancer hallmarks in oral and oro-pharyngeal carcinogenesis were identified, and interleukins were related to tumour-promoting inflammation and the tumour microenvironment in the pathogenesis [42]. Among numerous salivary interleukins (IL) studied in patients with OSCC and oral potentially malignant disorders, IL1β, IL6, and IL8, have been identified as significant biomarkers [20,36].

Due to variations in genetic, epigenetic, environmental, and habit-related risk factors associated with OSCC, geographical variations of salivary biomarker expression may be evident [37]. Sri Lanka presents a population with a high incidence of OSCC, attributed mainly to the betel quid chewing habit [6]. Despite this high incidence, salivary biomarkers are understudied in this population. To our knowledge, this is the first report on salivary interleukin levels in OSCC and OED in a Sri Lankan study cohort.

Our results revealed that salivary IL1β, IL6, and IL8 progressively increased from healthy, mild dysplasia, moderate dysplasia, severe dysplasia, and were at their highest levels in OSCC. These findings from a Sri Lankan study cohort align with findings from different populations around the world. Similar to our study, high levels of IL1β, IL6, and IL8 were reported in the saliva samples of patients with OSCC compared to healthy controls in a study conducted in Taiwan [43]. Another study in Taiwan concluded the same results for IL1β and IL8 [33]. Further, in studies conducted in Indian study cohorts, salivary IL1β, IL6 and IL8 appeared as significant biomarkers of OSCC and OPMD [30,44,45,46]. In studies conducted in USA, salivary IL6 and IL8 appeared as significant biomarkers of tongue OSCC [47], and as biomarkers that can monitor progression of oral lichen planus [48]. In a different study from USA, IL6 and IL8 were studied in saliva and serum, and salivary IL8 was significantly elevated in OSCC patients compared to controls [31]. Higher salivary IL6 and IL8 levels were also reported in patients with OSCC compared to other inflammatory oral diseases, including chronic periodontitis and oral lichen planus, in addition to healthy controls in a cohort from USA [32]. Several studies conducted in India reported similar results to ours, where IL1β and IL8 were significantly higher in OSCC and OPMD patients compared to controls, and were able to discriminate advanced-stage OSCC with significant power [30]. Salivary IL8 was considered an appropriate biomarker compared to the serum [46], and salivary IL6 was significantly elevated in OSCC and leukoplakia compared to controls [44]. Similar evidence was provided in studies conducted in Pakistan, Italy, Croatia, Poland and Hungary [24,27,37,49,50,51]. This overwhelming evidence from different populations is strongly validated, especially considering that all the studies measured interleukins in resting whole saliva, using similar techniques (e.g., ELISA and immune bead-based assays). However, it is important to highlight a few limitations of these studies, especially the absence of a uniform saliva collection protocol, variable mean biomarker values, and limited sample size. Further, there is scarce evidence from longitudinal cohort studies, importantly, with regards to fluctuation of salivary interleukin levels in the natural course of oral potentially malignant disorders. Overall, reported findings, together with the results of the current study, have significant translational potential, as quantitative evaluation of salivary ILs could aid to assess the risk of disease progression in OED and screening of OSCC.

In the present study, salivary IL6 and IL8 were able to discriminate OSCC from OED and controls with statistically significant results. Similar findings were reported by several studies, where the protein levels of salivary IL8 [52] and salivary IL6 [43] were capable of differentiating OSCC from controls. Similarly, mRNA of IL6 in saliva was capable of differentiating OSCC from controls [27]. These findings indicate that salivary IL6 and IL8 are potential biomarkers for OSCC screening.

The three ILs assessed in this study demonstrated to have several roles in carcinogenesis in different cancer types. In OSCC, IL1β is involved in the invasion and migration of cancer cells [53], while OSCC cells were reported to induce stromal cells to produce IL6, facilitating bone invasion by osteoclast formation [54]. In lung cancer, IL1β was reported to promote tumour genesis by facilitating inflammation, invasion, and angiogenesis [55]. IL8 has a role in promoting invasiveness and metastasis in breast cancer [56], and in the development of advanced colorectal cancer, through promoting and prolonging the inflammatory reactions in early-stage disease [57].

The sources of salivary IL6 in patients with OSCC were investigated using paired tumour tissues and saliva samples, where tumour tissue and tumour-infiltrating leukocytes were positively stained for IL6, indicating production of IL6 [27]. A difference in the expression of IL8 in the epithelium in patients with OSCC and OPMD was reported, where IL8 was present in all the layers of the epithelium in 64% of the OSCC group, but not in the control group, and no marked increase in expression in the OPMD tissue specimens was observed [24]. In a study investigating in vitro cytokine production, tumour-infiltrating lymphocytes produced a higher amount of IL1β and IL6 compared to peripheral blood leukocytes in OSCC patients [58]. These results indicate that higher levels of ILs observed in saliva in OSCC could be due to increased production from cancer cells and tumour-infiltrating leukocytes. However, ILs in saliva can also be derived from the systemic circulation, salivary gland tissue, and inflammatory cells not associated with cancer and non-cancer cells in the oral cavity, such as periodontal tissue. Studies to identify the authentic sources of IL6 and IL8, and mechanisms for their secretion into saliva, are future research needs.

Risk factors contribute to molecular changes that alter normal cellular functions. This promotes the transition of normal cells into malignancy. During this transformation, inflammatory responses play a crucial role. In our results, the protein levels of IL1β, IL6, and IL8 progressively increased from healthy, mild, moderate, and severe dysplasia, to OSCC. However, these biomarkers demonstrated variable associations with risk factors. This could be attributed to the small sample size that hinders accurate interpretation of the results.

Although many studies, including ours, indicate that salivary IL1β, IL6, and IL8 are promising biomarkers for OED and OSCC, this must be interpreted cautiously, as these are inflammatory cytokines. As such, the expression of these cytokines can increase with non-cancer-associated inflammatory conditions [59], such as periodontal disease [60,61]; although, it was reported that salivary levels of IL6 and IL8 in OSCC were not influenced by chronic periodontitis [32]. Strategies to overcome this include using a panel of biomarkers instead of a single biomarker, and establishing threshold values for OSCC, which need to be population- and risk-factor-specific [36].

The results of our study also indicate that salivary IL1β, IL6, and IL8 have limited ability as screening tools for OED, as the difference between OED and controls was not statistically significant. Singh and colleagues (2020) reported similar findings, where the discriminating ability of oral potentially malignant disorders using salivary IL1β and IL8 was not significant in a study cohort from India [30]. Conversely, significant differences in IL6 and IL8 biomarkers were reported in patients with oral potentially malignant disorders compared to controls in two other studies conducted in Poland and Croatia [24,51]. This could be related to the difference between the studied populations, in addition to the small sample size.

Our study was not without limitations, with the main limitation being the small sample size. We believe that true associations between risk factors and salivary interleukin levels could not be identified from the current dataset due to the limited number of samples in each risk-factor subcategory (e.g., users, ex-users and never users). We included all clinical stages and grades of OSCCs in one group. This limited the analysis of the variation of salivary IL levels in different clinical stages of OSCC. Therefore, larger studies are needed to confirm our findings.

## 5. Conclusions

Here we report that salivary protein levels of IL1β, IL6, and IL8 progressively increase from disease-free participants through different grades of OED, with the highest levels reported in OSCC, in a Sri Lankan study cohort. The studied salivary ILs have a significant translational potential to be developed into non-invasive, point-of-care tools for monitoring disease progression in OED, and screening and early detection of OSCC, applicable to low-resource settings. Population-specific and risk-factor-specific threshold values should be taken into consideration when translating salivary biomarkers into clinical diagnostic aids.

## Figures and Tables

**Figure 1 cancers-15-01510-f001:**
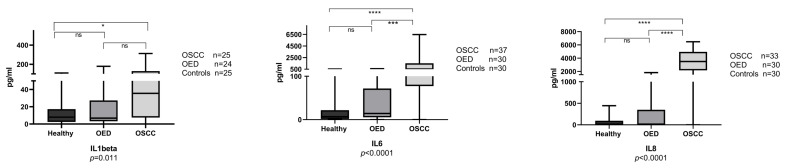
Salivary IL1β, IL6 and IL8 comparison in OSCC (oral squamous cell carcinoma), oral epithelial dysplasia (OED) and control groups. Box and whisker plots with bars for median and quartile strips, whiskers indicate minimum and maximum values for the salivary biomarker concentration in pg/mL. Statistical tests were the Kruskal-Wallis H test, and Dunn’s test for multiple comparisons among groups, n is the number of observations in each category. ns: no statistically significant difference in Dunn’s test (*p* > 0.05), asterisks indicate statistically significant differences for * *p*
< 0.05, *** *p*
< 0.001, **** *p*
< 0.0001.

**Figure 2 cancers-15-01510-f002:**
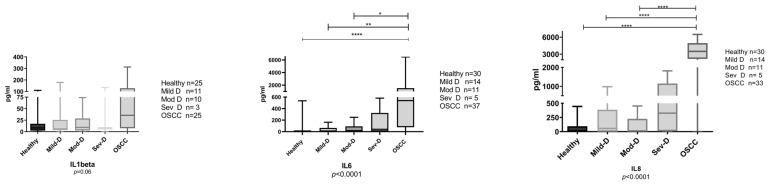
Salivary IL1β, IL6 and IL8 comparison in OSCC (oral squamous cell carcinoma), Mild-D (mild dysplasia), Mod-D (moderate dysplasia) and Sev-D (severe dysplasia) groups. Box and whisker plots with bars for median and quartile strips, whiskers indicate minimum and maximum values for the salivary biomarker concentration in pg/mL. Statistical tests were the Kruskal-Wallis H test, and Dunn’s test for multiple comparisons, n is the number of observations in each category, asterisks indicate statistically significant differences for * *p*
< 0.05, ** *p*
< 0.01, **** *p*
< 0.0001.

**Figure 3 cancers-15-01510-f003:**
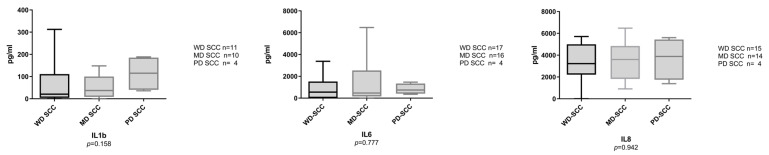
Salivary IL1β, IL6, and IL8 comparison in well differentiated (WD), moderately differentiated (MD), and poorly differentiated (PD) squamous cell carcinoma (SCC). Box and whisker plots with bars for median and quartile strips, whiskers indicate minimum and maximum values for the salivary biomarker concentration in pg/mL. Statistical tests were the Kruskal-Wallis H test, n is the number of observations in each category.

**Figure 4 cancers-15-01510-f004:**
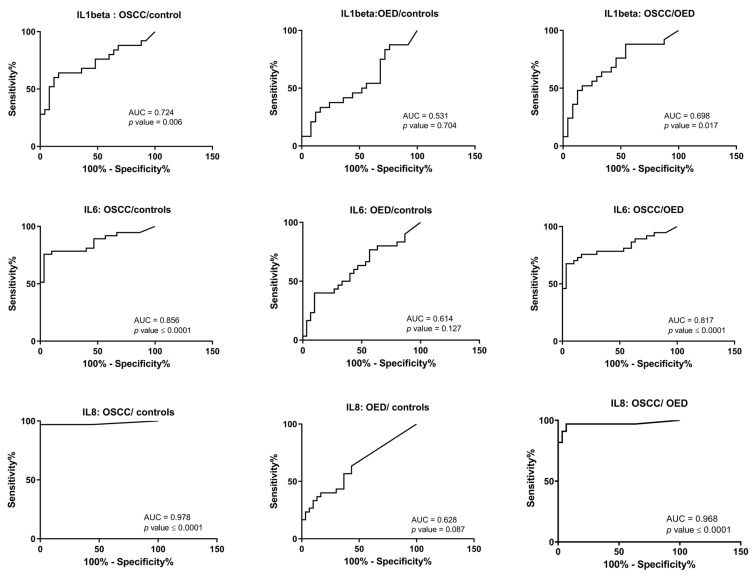
Receiver operating characteristic (ROC) curves for oral squamous cell carcinoma (OSCC) vs. controls, oral epithelial dysplasia (OED) vs. controls and OSCC vs. OED, for the three salivary interleukin biomarkers, IL1β, IL6, and IL8. AUC: area under the curve.

**Table 1 cancers-15-01510-t001:** Socio-demographic characteristics and risk factors of the study cohort.

		OSCC (n = 37)	OED (n = 30)	Controls (n = 30)
Age (years)	Mean Range	60.0 ± 11.5 33–78	56.2 ± 12.3 29–80	62.2 ± 10.2 36–79
Income (LKR)	Mean Range	24,875 ± 19,766 1000–70,000	26,782 ± 17,474 5000–60,000	40,533 ± 10,953 25,000–60,000
Sex	Male (%) Female	31 (84%) 6 (16%)	24 (80%) 6 (20%)	24 (80%) 6 (20%)
Ethnicity	Sinhala (%) Other	30 (81%) 7 (9%)	28 (93%) 2 (7%)	30 (100%) -
Smoking	Daily Never Ex-user	12 (32%) 20 (54%) 5 (14%)	10 (33%) 16 (54%) 4 (13%)	6 (20%) 15 (50%) 9 (30%)
Alcohol	Daily Never Ex-user	23 (62%) 12 (33%) 2 (5%)	13 (44%) 15 (50%) 2 (6%)	11 (37%) 16 (53%) 3 (10%)
Betel quid	Daily Never Ex-user	25 (67%) 11 (30%) 1 (3%)	17 (57%) 6 (20%) 7 (23%)	10 (34%) 18 (60%) 2 (6%)
Family history of any cancer type	Yes No	9 (24%) 28 (76%)	6 (20%) 24 (80%)	5 (17%) 83 (83%)
Mouthwash use	Yes No	3 (8%) 34 (92%)	9 (30%) 21 (70%)	1 (4%) 29 (96%)
Co-morbidity	Yes No	8 (22%) 29 (78%)	16 (53%) 14 (47%)	15 (50%) 15 (50%)

LKR: Sri Lankan rupees.

**Table 2 cancers-15-01510-t002:** Anatomical location of lesions in patients with OSCC and OED.

	OSCC (n = 37)	OED (n = 30)
Lips/labial mucosa	1 (2.8%)	2 (6.6%)
Buccal mucosa	14 (37.8%)	24 (80.2%)
Tongue	14 (37.8%)	2 (6.6%)
Gingiva	1 (2.8%)	-
Floor of the mouth	3 (8.4%)	-
Palate	2 (5.6%)	2 (6.6%)
Alveolus/bone	2 (5.6%)	-

**Table 3 cancers-15-01510-t003:** Discriminating ability of the salivary biomarkers using receiver operating characteristic (ROC) analysis.

		OSCC/Controls	OED/Controls	OSCC/OED
IL1β	AUC	0.724	0.531	0.698
	95% CI for AUC	0.579–0.870	0.366–0.697	0.549–0.847
	*p*-value	0.006	0.704	0.017
	Cut-off value	>20.49 pg/mL	-	>33.4 pg/mL
	Sensitivity at COV	64%	-	52%
	Specificity at COV	84%	-	83.3%
IL6	AUC	0.856	0.614	0.817
	95% CI for AUC	0.763–0.950	0.470–0.758	0.712–0.922
	*p*-value	<0.0001	0.127	<0.0001
	Cut-off value	>95.9 pg/mL	-	>169.7 pg/mL
	Sensitivity at COV	75.6%	-	70.2%
	Specificity at COV	96.6%	-	90%
IL8	AUC	0.978	0.628	0.968
	95% CI for AUC	0.934–1.02	0.486–0.769	0.918–1.01
	*p*-value	<0.0001	0.087	<0.0001
	Cut-off value	>394.3 pg/mL	-	>1067 pg/mL
	Sensitivity at COV	96.9%	-	90.9%
	Specificity at COV	96.7%	-	96.6%

The AUC is the area under the curve of the receiver operating characteristics curve (ROC). CI is the confidence interval, COV is the cut-off value, the significant discriminating ability was denoted for *p*-values less than 0.05 and AUC values more than 0.7, COV was calculated for significant AUCs only.

## Data Availability

Raw data and additional statistical analyses are provided in the Appendix A associated with this manuscript.

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
