# Peer review of "Salivary Interleukin Levels in Oral Squamous Cell Carcinoma and Oral Epithelial Dysplasia: Findings from a Sri Lankan Study"

_cancers, 2023, doi:10.3390/cancers15051510_

Round 1

Reviewer 1 Report (Previous Reviewer 3)

authors have considered suggestions

Author Response

We would like to thank the reviewer for taking the time to look at our manuscript again and for confirming we have addressed all their comments and suggestions.

Reviewer 2 Report (Previous Reviewer 2)

The Authors have referred to all the comments of the Reviewers from the previous submission and improved the manuscript considerably.

However, I have a few comments to revise the Results section, in particular the plots. The figures should have a better resolution. Also, please expand the used abbreviations in the captions. For ROC curves please correct the axes so that they end at 100 instead of 150. In the same charts, correct the phrase "p-value =<" to "p-value <". Standardise the term “p-value” in all diagrams.

Author Response

Comment: The Authors have referred to all the comments of the Reviewers from the previous submission and improved the manuscript considerably.

Response:

We would like to thank the reviewer for taking the time to look at our manuscript and for confirming we have addressed all their comments and suggestions.

Comment: I have a few comments to revise the Results section, in particular the plots. The figures should have a better resolution.

Response:

All the figures were submitted in TIF (tagged image file) format with 1200 dpi resolution, in addition to the embedded figures in the word file.

Comment: Also, please expand the used abbreviations in the captions.

Response:

All abbreviations in the figure captions were expanded.

Comment: For ROC curves please correct the axes so that they end at 100 instead of 150.

Response:

All ROC curves we modified so that Y axes end at 100.

Comment: In the same charts, correct the phrase "p-value =<" to "p-value <". Standardise the term “p-value” in all diagrams.

Response:

The term ‘p-value’ was standardized throughout the manuscript, including figures and tables (highlighted in yellow).

This manuscript is a resubmission of an earlier submission. The following is a list of the peer review reports and author responses from that submission.

Round 1

Reviewer 1 Report

Dear authors,

Thank you for submitting this manuscript. However, the study of salivary interleukins in patients with OSCC and oral dysplasia has already been conducted many times, as evidenced in your discussion and the papers you have cited.

Since the patients you are measuring salivary IL on already have visible OSCC or dysplasia in areas easily visible to the naked eye - what is the added benefit to measuring salivary IL ? It cannot be for diagnostic purposes because you can already clearly see the carcinoma on the lip or tongue or other oral cavity areas which are in contact with the saliva.

Unless you have discovered a mechanism through which the IL are produced in these carcinomas then there is nothing novel in this current manuscript

Reviewer 2 Report

The topic of the manuscript is to investigate the levels of IL1β, IL6 and IL8 in saliva in disease free controls and patients with OSCC and OED, and their associations with risk factors in a Sri Lankan study cohort.

The title and the abstract of the article are informative. The Introduction clearly presents the issue of oral squamous cell carcinoma, as well as interleukins and their role in carcinogenesis. The section "Material and Methods" precisely describes the chosen study design, however, the statistical subsection should be modified. The section "Results" should be revised from the statistical side. The Discussion is interestingly written, however, the paragraph about the study limitations could be improved and the more recent references should be supplemented. The Conclusions seem to be the "take-home" messages.

Some following points must be clarified/corrected for the further processing of this article.

Merits-related comments:

1.       In the abstract, please indicate the number of patients in subgroups.

2.       In the description of the statistical methods, change the sentence about the comparisons. The ROC analysis is not used to compare but to assess the differentiation/discrimination of subgroups. The significance of the ROC curves is also determined by p-value rather than by a specific value of AUC.

3.       Which method was used to determine the cut-off in ROC analysis? Please complete in the methodology section.

4.       Since the authors used the Kruskal-Wallis test, the variables did not correspond to the normal distribution (which should be mentioned in the methodology). Therefore, the results should be presented in the form of box plots with medians and quartile stripes (and not with mean and standard errors).

5.       Please supplement the Results section with a graphical representation of the ROC curves – three interleukins together in each figure but separate diagrams for three distinctions.

6.       The study limitations should be extended at the end of the Discussion.

7.       It is suggested to add more recent articles from 2020-2022 to the references in the Introduction and the Discussion.

8.       In the Introduction or the Discussion, the Authors could mention recent innovative reports on the early detection of oral squamous cell carcinomas using saliva, e. g. its metabolites (useful references: 10.3390/metabo12040294, 10.3390/metabo11090587).

Technical comments:

1.       The manuscript requires editorial editing, e. g. typing errors (in lines 125-126: “C0”).

2.       The abstract should be a single paragraph and should follow the style of structured abstracts, but without headings.

3.       Please use “p-value” instead “p”.

4.       References should be described as follows:
1. Author 1, A.B.; Author 2, C.D. Title of the article. 
Abbreviated Journal Name YearVolume, page range.

Reviewer 3 Report

The topic is important and timely. Considering that the topic refers to oral cavity (pre)malignant lesions, treatment strategies, as well as treatment toxicities and advances in research should be better described. Therefore, introduction/discussion would be enhanced by addition of PMID: 27933385, PMID: 30952736, PMID: 31683170 references to better contextualize the issue at hand in oncologic scenario. Limits and strengths of the study supporting evidence and validity of the research should be stressed in the discussion.